# Aerobic Glycolysis: A DeOxymoron of (Neuro)Biology

**DOI:** 10.3390/metabo12010072

**Published:** 2022-01-13

**Authors:** Avital Schurr, Salvatore Passarella

**Affiliations:** 1Department of Anesthesiology and Perioperative Medicine, University of Louisville School of Medicine, Louisville, KY 40202, USA; 2Department of Biomedical Sciences and Human Oncology, University of Bari “Aldo Moro”, 70124 Bari, Italy; spassarella3@gmail.com

**Keywords:** aerobic glycolysis, astroglial-neuronal L-lactate shuttle, BOLD fMRI (blood oxygen level-dependent functional magnetic resonance imaging), CMR (cerebral metabolic rate), glucose, L-lactate, mitochondrial oxidative phosphorylation, oxygen

## Abstract

The term ‘aerobic glycolysis’ has been in use ever since Warburg conducted his research on cancer cells’ proliferation and discovered that cells use glycolysis to produce adenosine triphosphate (ATP) rather than the more efficient oxidative phosphorylation (oxphos) pathway, despite an abundance of oxygen. When measurements of glucose and oxygen utilization by activated neural tissue indicated that glucose was consumed without an accompanied oxygen consumption, the investigators who performed those measurements also termed their discovery ‘aerobic glycolysis’. Red blood cells do not contain mitochondria and, therefore, produce their energy needs via glycolysis alone. Other processes within the central nervous system (CNS) and additional organs and tissues (heart, muscle, and so on), such as ion pumps, are also known to utilize glycolysis only for the production of ATP necessary to support their function. Unfortunately, the phenomenon of ‘aerobic glycolysis’ is an enigma wherever it is encountered, thus several hypotheses have been produced in attempts to explain it; that is, whether it occurs in cancer cells, in activated neural tissue, or during postprandial or exercise metabolism. Here, it is argued that, where the phenomenon in neural tissue is concerned, the prefix ‘aerobic’ in the term ‘aerobic glycolysis’ should be removed. Data collected over the past three decades indicate that L-lactate, the end product of the glycolytic pathway, plays an essential role in brain energy metabolism, justifying the elimination of the prefix ‘aerobic’. Similar justification is probably appropriate for other tissues as well.

## 1. Introduction

The terms aerobic and anaerobic metabolism have their roots in studies of bacterial fermentation and later skeletal muscle. ‘Aerobic glycolysis’, a term borrowed from a phenomenon typical in proliferating cancer cells [1], was made an accepted biochemical term of brain metabolic activity by Fox et al. [2]. It describes an observation in which a visual stimulation caused an approximately 50% increase in the cerebral metabolic rate of glucose (CMR_glc_) compared with an approximately 5% increase in the cerebral metabolic rate of oxygen (CMRO_2_). In other words, the increase in the rate of glucose utilization upon neuronal activation was not matched by a similar increase in oxygen utilization. Accordingly, of the two principal substrates of ATP production, glucose and oxygen, mainly glucose was utilized without a matching utilization of oxygen. The investigators concluded that the less efficient pathway of energy metabolism, i.e., glycolysis, was activated by the visual stimulation they used, bypassing the much more efficient pathway with which it is normally coupled, the mitochondrial oxphos and thus ‘aerobic glycolysis’. Although this phenomenon is accepted today by many as real, a logical explanation for this thermodynamically defying phenomenon has been missing. Explaining it is of paramount importance, as the methodology used to originally establish its existence has developed into what is considered today’s mainstay of neuroimaging (fMRI) combined with blood oxygen level-dependent (BOLD) signal employed for the calculation of CMRO_2_. After all, measuring function-dependent brain consumption of both glucose and oxygen allows us to study myriad cognitive and behavioral functions as they relate to their energy needs, as exemplified by studies on high intensity exercise and executive function [3,4]. Recently, Theriault et al. [5] offered a hypothesis in an attempt to resolve the metabolic mystery of ‘aerobic glycolysis’, according to which there is an efficiency tradeoff that explains it. Several assumptions are at the basis of their hypothesis:

**Assumption** **A1.***Blood oxygen level directly reflects the oxygen level in the surrounding neural tissue itself. If, upon stimulation, blood oxygen level does not drop, neither does the oxygen level in the stimulated neural tissue*.

**Assumption** **A2.***There must be a cellular mechanism that allows glycolysis, despite the presence of ample amount of oxygen, to proceed to its final anaerobic metabolic fate, L-lactate, bypassing the mitochondrial oxphos. Therefore, according to this hypothesis, glycolysis somehow does not end with pyruvate, the principal substrate of mitochondrial oxphos, but with L-lactate. Hence, this hypothesis relies on the lingering old dogma, where there are two separate outcomes to glycolysis, aerobic, which ends with pyruvate, and anaerobic, which ends with L-lactate*.

**Assumption** **A3.***The glycolytic pathway is faster and under tighter control than the mitochondrial oxphos. Glycolysis is quick to provide the amount of ATP necessary to support the required activity induced by a given stimulus, while mitochondrial oxphos is slower to do so and produces ATP at amounts above those required, creating an ATP storage problem and possible waste*.

Provided here is a consideration of the ‘aerobic glycolysis’ phenomenon in light of published data that could prove the phenomenon to be a simple misconception at best, or a misleading concept at worst. Moreover, this phenomenon could provide the necessary evidence to persuade the remaining doubters that L-lactate is always the final end product of glycolysis and the main substrate of oxphos. This consideration also supports the astrocyte-neuronal L-lactate shuttle (ANLS) hypothesis.

## 2. Functional Brain Energy Metabolism

At the end of the nineteenth century, the brain was believed to be an organ without much activity [6]. Then, in 1913, Tashiro [7] demonstrated that nerve tissue produces CO_2_. Efforts to quantify energy metabolism with brain activity had started in earnest in the mid 1950s and gained momentum with the seminal work of the Sokoloff group [8,9], when they put to use the [^14^C]deoxyglucose method for the quantitative measurement of local cerebral glucose consumption. By 1986, the Raichle group combined oxygen extraction measurement using ^15^O-labled radiotracers and positron emission tomography (PET) to demonstrate an uncoupling between cerebral blood flow (CBF) and local cerebral metabolic rate of oxygen (CMRO_2_) in response to neuronal activation induced by somatosensory stimulation [10]. Two years later, the same group [2] added to their measurements brain glucose uptake. They concluded that “Transient increases in neural activity cause a tissue uptake of glucose in excess of that consumed by oxidative metabolism, acutely consume much less energy than previously believed, and regulate local blood flow for purposes other than oxidative metabolism.” The nonoxidative glucose consumption during focal physiologic neural activity, referred to as ‘aerobic glycolysis’, suggested that glucose metabolism is confined to glycolysis alone, despite the presence of ample amounts of oxygen in the blood. This unexpected finding was confirmed by others [11,12,13,14,15,16]. It is worth mentioning here that the term ‘aerobic glycolysis’ was actually used by Otto Warburg in connection with cancer cells, known as the Warburg Effect [1], as well as to describe the process that provides the necessary ATP for the function of the Na^+^/K^+^ATPase pump [17,18]. In contrast to studies that reported ‘aerobic glycolysis’ upon neural stimulation, other studies produced results that specifically contradict this phenomenon. These studies all demonstrate that, upon stimulation, glucose uptake is accompanied by oxygen uptake, i.e., oxidative utilization of glucose [19,20,21,22,23,24,25,26,27]. There is a general agreement among investigators that neither cerebral blood flow (CBF) measurements nor BOLD contrast analyses are reliable enough methods to measure CMRO_2_ and, therefore, models were offered to deal with this unreliability [28,29,30]. While the purpose of this manuscript is not to debate the advantages and disadvantages of BOLD or any other indirect measurement of CMRO_2_, one must be careful when using data produced by these methods to arrive at far reaching conclusions about the cellular, biochemical, and molecular processes of functional brain energy metabolism. It is somewhat surprising that Theriault et al. [5] decided to put forward an all-encompassing hypothesis about the efficiency tradeoff of ‘aerobic glycolysis’. This is surprising because the authors themselves recognize the drawbacks of BOLD fMRI measurements. 

Although it seems that all the investigators delving into this particular research endeavor agree that measuring CMRO_2_ indirectly has its own drawbacks, and despite the fact that direct measurements of CMRO_2_ are available, one is pushed hard to find published studies where both CMR_glc_ and CMRO_2_ are measured directly and simultaneously in cerebral tissue. Oxygen electrodes to measure local tissue oxygen levels in vivo have been constructed and used for decades. Similarly, electrodes/biosensors that can measure the tissue levels of glucose are also available. Only one study wherein both brain tissue oxygen and glucose levels were measured directly in vivo at rest and during activation was published [21]. This study has received over 330 citations. In addition, the investigators also used a L-lactate electrode/biosensor to measure the fluctuations of L-lactate levels simultaneously with those of glucose and oxygen. The data embedded in that study provide the most unique picture of the interrelationship between the three most important substrates of energy metabolism in the brain; any guesswork or assumption in interpreting the results is eliminated, as the data were acquired via direct measurements. Obviously, the use of such biosensors is limited only to animal, not human, research. Previous analyses of that study’s results were performed and published earlier [31,32]. Nevertheless, as that study still stands alone, many of the following arguments justify revisiting it and reanalyzing its results, while taking into consideration that the use of electrodes has its limitations as the fluctuations in the levels of glucose, lactate, and oxygen do not necessarily reflect just utilization or the lack of it. They, for instance, could indicate uptake of glucose and/or lactate through upregulation of their respective transporters [33].

## 3. ‘Aerobic Glycolysis’ Is a Paradox

In their hypothesis paper, Theriault et al. [5] describe ‘aerobic glycolysis’ as a paradox, as this phenomenon is illogical, when common knowledge and understanding of the process of oxidative energy metabolism are considered. It is also paradoxical that the authors have chosen to offer a logical explanation for a phenomenon whose existence was disputed by other studies that actually did measure oxygen uptake and utilization upon neural activation [19,20,21,22,23,24,25,26,27]. Even if, for the sake of argument, the existence of ‘aerobic glycolysis’, i.e., the conversion of glucose to L-lactate in the presence of oxygen without any participation of oxphos upon stimulation, is accepted as valid, the biochemical mechanism that allows the ‘aerobic glycolysis’ to produce L-lactate, rather than pyruvate, supposedly the substrate of mitochondrial oxphos according to the old school, is left unexplained. The scenario wherein pyruvate is the end product of aerobic glycolysis was formulated when the glycolytic pathway was elucidated eight decades ago. Nevertheless, the part of the hypothesis that assumes ‘aerobic glycolysis’ to end up with lactate is in full agreement with the claim that L-lactate is always the end product of the glycolytic pathway under both anaerobic and aerobic conditions and, under the latter, the substrate of oxphos [34,35,36,37,38]. A recent review summarizes the overwhelming evidence supporting the existence of a mitochondrial lactate dehydrogenase (mLDH), which is responsible for the conversion of L-lactate to pyruvate intramitochondrially [38]. L-lactate formation upon neural activation (stimulation) was clearly recorded by Hu and Wilson [21] (Figure 1); these results have been analyzed and discussed previously [31,32]. However, in contrast to the assumption by the aforementioned hypothesis, L-lactate was formed from glucose and, more importantly, its utilization was accompanied by oxygen consumption with every stimulation, i.e., L-lactate was the substrate of the mitochondrial oxphos [21]. That study measured an average of 0.13 mM O_2_ consumption with each stimulation and the small tissue fluctuations measured in O_2_ strongly suggest that O_2_ supply is not limited during activation, a fact that agrees with the model offered by Buxton et al. [28] and Hyder et al. [39].

This amount of O_2_ would oxidize 0.043 mM of L-lactate in the mitochondrial oxphos to yield an average of 0.65 mM ATP. In addition, Hu and Wilson measured an average consumption of glucose upon each stimulation of 0.094 mM (based on an in vivo glucose baseline concentration of 2.6 mM). Consumption of glucose of 0.094 mM would have produced only 0.19 mM ATP via glycolysis. This calculation is based on the fact that L-lactate oxidative utilization is 15-fold more efficient than glycolytic glucose utilization, where ATP production is concerned. Moreover, Hu and Wilson [21], along with many others [35,40,41,42,43,44,45,46,47,48,49,50,51], have shown a significant increase in brain L-lactate upon stimulation or physical exercise. Interestingly, the performance of magnetic resonance spectroscopy (^1^H-MRS) after exhaustive exercise showed that the cerebral arterio-venous difference for glucose (AV_glc_) increased from 0.6 mM at rest to 0.9 mM at the point of exhaustion, whereas AV_lac_ increased from 0 to 1.3 mM and AV_O__2_ increased from 3.2 to 3.5 mM [44]. Maximal exercise reduced the cerebral metabolic ratio (O_2_/glucose; respiratory quotient) from 6.0 ± 0.3 to 2.8 ± 02 and it remained low during early recovery. Such a reduction can be easily explained when one considers that the cerebral metabolic ratio [O_2_/L-lactate] = 3. The simple explanation for these investigators’ findings is that, at maximal exercise, L-lactate completely or nearly completely replaced glucose as the oxidative cerebral energy substrate. This is not so for those who still insist that glucose is always the main substrate for energy metabolism in the brain. Even when L-lactate is shown to replace glucose as the main oxidative substrate, there is an insistence by some to include glucose as part of the equation (O_2_/[glucose + 1/2 L-lactate]) and to conclude that carbohydrate is always the brain’s principal fuel [5,52]. Of course, glucose is the substrate of glycolysis and the prime source of L-lactate. However, given that there is an ample supply of the latter to the brain, including from skeletal muscles, L-lactate will be the preferential energy substrate for oxphos, bypassing glucose. As long as there is an (unjustified) objection to accept L-lactate as the end-product of glycolysis, either in the presence or absence of oxygen, and as the main oxidative substrate of mitochondrial oxphos, ‘aerobic glycolysis’ will remain a ‘paradox’. What mechanism, if any, determines when glycolysis ends with pyruvate (the presumed end-product of the ‘real’ dogmatic aerobic glycolysis) or with L-lactate (the end-product of the ‘aerobic glycolysis)? If one accepts the fact that cerebral glycolysis during stimulation proceeds through L-lactate and stops there, while avoiding the mitochondrial oxphos, why can glycolysis not proceed to end with lactate also at rest? A great deal of the confusion brought about by the term ‘aerobic glycolysis’ could be avoided if glycolysis and oxphos were accepted as two independent biochemical pathways—the former with glucose as its substrate and lactate as its end product, and the latter with lactate as its substrate and CO_2_ and H_2_O as its end products. As long as the doctrinaire concept exists regarding the glycolytic pathway having two separate endings, one when oxygen is present (pyruvate) and another when oxygen is absent (L-lactate), the paradoxical phenomenon of ‘aerobic glycolysis’ will continue to look for an explanation. As discussed in the following section, an aerobic glycolysis, the conversion of glucose to L-lactate in the presence of oxygen, exists not only in tissues, such as red blood cells, but also in brain tissue, neurons, glia, axons, and dendrites, where ion pumps, and especially the Na^+^/K^+^ATPase pump, have their own exclusively devoted glycolytic pathway that produces ATP for their pumping needs and where L-lactate is the end product [17,18,53].

## 4. Astroglial Glycolysis and Neuronal Oxphos—The Lactate Shuttle at Work

At the basis of the efficiency tradeoff hypothesis of ‘aerobic glycolysis’ is the assumption that glycolysis itself, a primitive metabolic pathway for the inefficient production of ATP, offers a better solution for providing the necessary energy required by neurons, synapses, and axons during activation. This assumption is a direct outcome of accepting the premise, forwarded since the mid 1980s, that certain modes of cerebral activation are supported by glucose consumption not accompanied by oxygen consumption [2,10]. However, many studies disagree with the idea of ‘aerobic glycolysis’, as cerebral oxygen consumption upon activation has been detected. Among them, the study by Hu and Wilson [21] paints a clear picture of activated neural tissue in vivo that consumes glucose, oxygen, and lactate (Figure 1). Actually, after the application of the very first stimulation in that study, where the most glucose was consumed, along with oxygen and lactate, from the 2nd and through the 10th stimulation, lactate levels increased several fold and, with each stimulation, L-lactate became the principal substrate to be consumed along with oxygen. L-lactate production appeared to increase continuously [31,32] up to the last stimulation. Although L-lactate production upon neural stimulation (activation) has been observed by many, when such production is accompanied by oxygen consumption, a consideration must be given to the astroglial-neuronal L-lactate shuttle (ANLS) hypothesis, first offered by Pellerin and Magistretti [54]. This hypothesis postulates that glutamate, released upon neural stimulation and taken up by astrocytes, activates glycolysis, which produces and releases L-lactate that is utilized aerobically by neurons. Pellerin and Magistretti described this L-lactate production by astrocytes as ‘aerobic glycolysis’, a term that thematically may be correct, as lactate is produced despite the presence of oxygen, but it is clearly confusing and misleading considering the fact that the L-lactate thus produced is, concomitantly, oxidized. Interestingly, it has been shown that mitochondria from both prostate and HepG2 cancer cells can take up and metabolize L-lactate. Moreover, pyruvate transport does not occur in cancer cell mitochondria [55,56,57]. Such metabolism clearly indicates that cancer cells, where ‘aerobic glycolysis’ was first observed, do use oxygen and lactate to produce mitochondrial ATP. If one considers the proposal that L-lactate is always the end-product of glycolysis, both aerobically and anaerobically, then ‘aerobic glycolysis’ is a term that should not be used. Of course, this confusion traces back to the early days when the glycolytic pathway and its several enzymatic steps were first elucidated and when pyruvate and L-lactate as end products were offered as two completely separate outcomes of the pathway, aerobic and anaerobic, respectively [32]. Nevertheless, the study by Hu and Wilson [21] clearly demonstrates the ANLS in action. Neural activation stimulates glucose uptake (by astrocytes?), followed immediately by L-lactate production and its oxidative consumption (by neurons?). Moreover, upon cessation of activation, the accumulated L-lactate was slowly consumed over time, while the glucose level increased above the baseline level, indicating that, during rest, when L-lactate is abundant, it is the preferred energy substrate over glucose [34,35,40,42,43,44,45,46,47,48,50,58,59,60], a phenomenon that can be described as glucose sparing [61]. The ANLS hypothesis’ main requirement is that astrocytes produce via ‘aerobic glycolysis’ the necessary L-lactate to be shuttled into neurons. In their original publication of their hypothesis, Pellerin and Magistretti [54] postulated that this glycolytic supply of ATP is necessary to support the Na^+^/K^+^ATPase pump, which is activated by the excitatory neurotransmitter, glutamate. It is well established that the plasma membrane-bound Na^+^/K^+^ATPase pump requires glycolysis to supply the necessary ATP for its function [17,18,53]. It is a system devoted to this particular function, and hence ‘aerobic glycolysis’. As such, the pump’s energy supply does not have an alternative source of ATP (oxphos), because of either structural or localization constraints. If the Na^+^/K^+^ATPase pump is responsible for the majority of the L-lactate production upon excitation (glutamatergic), one could predict that L-lactate is not only transported (via the monocarboxylate transporters, MCTs) into neurons for their own oxphos, but is also utilized by the astrocytes themselves for the same purpose. A similar principle is at work in skeletal muscle and in the heart [62,63,64]. Therefore, if the majority of ‘aerobic glycolysis’ reported in studies using BOLD fMRI is due to glycolytic glucose consumption that supports ion pumps such as the Na^+^/K^+^ATPase [65], then it would be meshed neatly with the observations by Hu and Wilson [21] and with the ANLS hypothesis [54], where glycolytic glucose consumption, the production of L-lactate, and its oxidative consumption all work in concert. This scenario would account for the metabolic events that follow cerebral activation: (1) increased CMR_glc_ via glycolytic ATP production to support the Na^+^/K^+^ATPase pump; (2) increased L-lactate production due to this heightened glycolytic activity; and (3) decreased CMRO_2_ from a respiratory quotient of ~6 to ~3, the expected quotient of the reaction CH_3_CHOHCO_2_H + 3O_2_ → 3CO_2_ + 3H_2_O.

As part of their hypothesis, Theriault et al. [5] suggest that ‘aerobic glycolysis’ takes place in thin axons that are too thin to contain mitochondria. While it was suggested that, the thinner the axon, the fewer mitochondria it contains [66,67], mitochondria are known to be mobilized within the axon [68], and reports show that axons without mitochondria cannot survive [69]. Nevertheless, thin axons, similar to neurons, astroglia, synapses, and thick axons, likely contribute to glycolytic glucose utilization and L-lactate production upon activation, as they are all equipped with ion pumps. Additionally, this hypothesis assumes that ATP supply is better controlled and easier to turn on and off with ‘aerobic glycolysis’ than oxphos. Upon an initial stimulation, glycolysis is clearly the first pathway to respond to the need for increased ATP supply; however, as can be seen from the study by Hu and Wilson [21], along with the increase in glucose consumption, there is an increase in the consumption of both L-lactate and oxygen. Following the first stimulation and with every additional one, less and less glucose is consumed, while the production and oxidative consumption of L-lactate increased and persisted. A study using both cell cultures and organo-typical hippocampal slices has demonstrated that a low mM concentration of K^+^ increased astrocytic glycolysis by fourfold within seconds, an effect that was immediately ceased upon removal of this agonist [65]. In contrast, glutamate stimulation persisted unabated for >20 min. Both K^+^ and glutamate stimulations required an active Na^+^/K^+^-ATPase pump and both modulate astrocytic glycolysis, a short-term modulation by the former and a long-term one by the latter, which leaves a metabolic trace. These results also support the role assigned to astrocytic L-lactate production and its oxidative neuronal utilization by the ANLS hypothesis [54], and are in complete agreement with the findings of Hu and Wilson [21]. In short, once a stimulation occurred and glycolysis is initiated, further activation of neural tissue is accompanied by L-lactate oxidative metabolism via oxphos, which provides the majority of the necessary ATP for the most demanding function i.e., pumping ions, and especially the Na^+^/K^+^-ATPase activity. These findings argue against the claim that ‘aerobic glycolysis’ is somehow an efficiency tradeoff that allows for faster response and better control over ATP supplies upon neural activation. Clearly, efficiency is absolutely an important consideration when a response to ATP demand is needed, as L-lactate becomes a major substrate of oxphos during and post-activation of neural tissue. Of course, glucose is the source of L-lactate and glycolysis is the pathway necessary to provide it. However, once produced, along with its existing levels within the neural tissue and its additional supplies from skeletal muscles and other organs and tissues, L-lactate does not appear to be in short supply. When all is said and done, at the end of the activation period, the accumulated L-lactate is still the preferred substrate for ATP production over glucose [21]. As to the issue of stricter control over the production of ATP by glycolysis in comparison with oxphos, and that somehow the latter may produce a flood of ATP that would create a problem of storage shortage or waste, there is no evidence for such a scenario. Actually, oxphos is given to more points throughout its pathway where its rate can be controlled [70]. Moreover, again from the study of Hu and Wilson [21], it is clear that an abundance of L-lactate tilts the production of ATP towards oxphos and away from glycolysis, indicating product inhibition of the latter. Based on the analysis of these results, repeated stimulation drove both the production of L-lactate (Figure 1) and its utilization (Figure 2) up, while oxygen utilization remained more or less constant. The increase in L-lactate utilization above and beyond the amount utilized with oxygen could indicate that the extra L-lactate is being used for glycogen buildup. Nevertheless, the high level of L-lactate produced during those repeated stimulations indicates that this monocarboxylate is the preferred energy substrate. This was also observed during the recovery period, where inhibition of glucose utilization can be clearly seen as the L-lactate level slowly diminished over time, while glucose accumulated above the baseline level, as it remained unused (Figure 1).

## 5. Summary

The time has arrived to drop the words ‘aerobic’ and ‘anaerobic’ when dealing with glycolysis. These are archaic terms that originated more than a century ago, when L-lactate was considered to be a waste product of glycolysis that the tissue must be rid of. Combining it with the erroneous claim that, in the presence of oxygen, glycolysis ends up with pyruvate, the presumed substrate of oxphos, led to the formulation of two distinct glycolytic pathways, aerobic and anaerobic. Ample evidence is now available to show that glycolysis always ends with L-lactate, not with pyruvate. Therefore, the presence of oxygen does not change glycolysis’ end product from L-lactate to pyruvate, it only allows L-lactate to become the substrate of the mitochondrial oxphos. When oxygen is absent, L-lactate, the end-product of glycolysis, accumulates.

There is a persistent tendency among many to continue the use of outdated terms that no longer fit recent data and knowledge, a tendency that could be attributed, among others, to what was coined “habit of mind” [36,71]. The misuse of scientific terms such as “heavy metals” [72] has implications beyond the simple scientific inaccuracy [73]. Therefore, efforts must be made to avoid at all cost the use of outdated, obsolete, or misused scientific terms.

## Figures and Tables

**Figure 1 metabolites-12-00072-f001:**
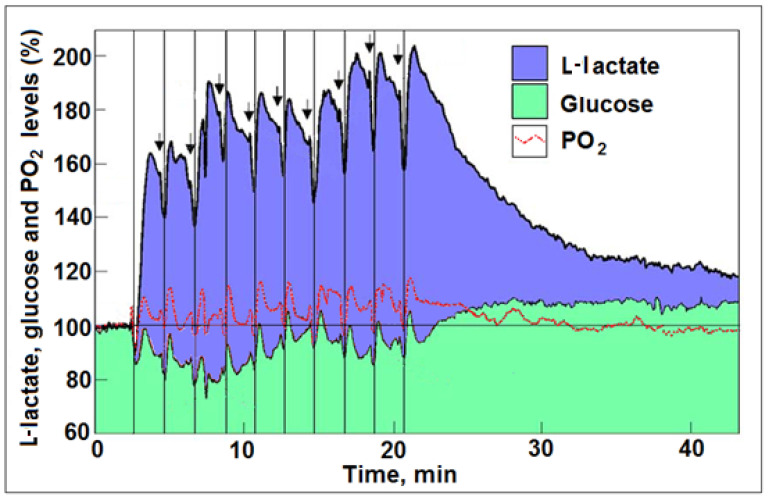
Profiles of time course and dynamic relationships of local extracellular L-lactate, glucose, and PO_2_ levels in the rat hippocampal dentate gyrus during a series of 5 s electrical stimulations (arrows) of the perforant pathway at 2 min rest intervals (reproduced from [31]). The changes in the mean concentration of glucose were always in the opposite direction to the changes in mean L-lactate concentration. The vertical lines were drawn to indicate the simultaneous dip in all three analytes in response to each of the electrical stimulations. For additional details, see [31].

**Figure 2 metabolites-12-00072-f002:**
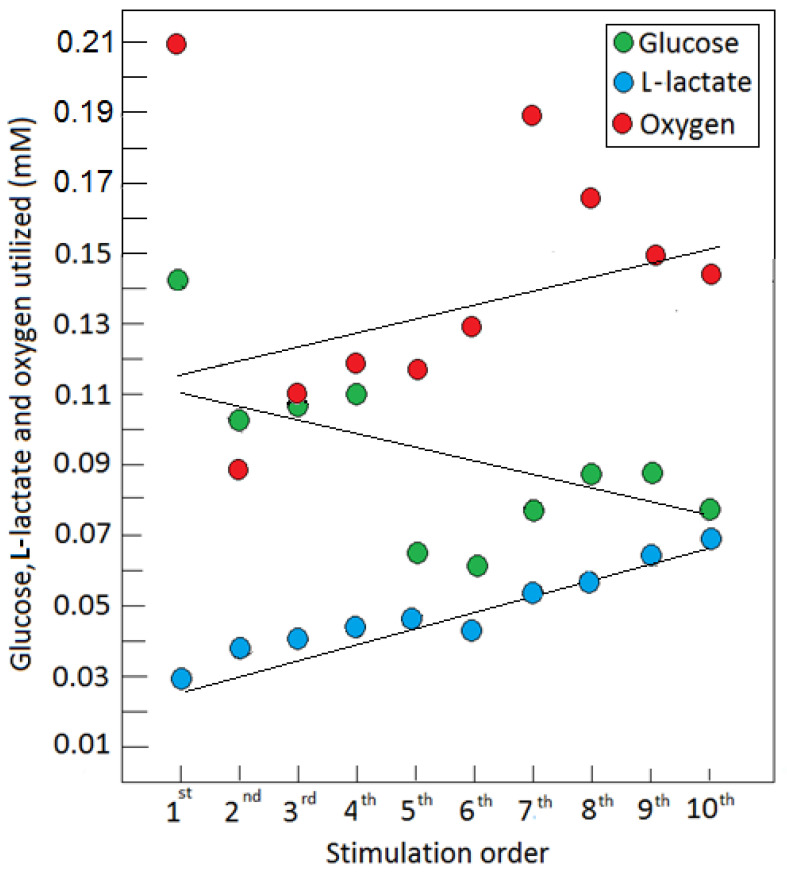
The amount, in mM, of each of the three substrates of cerebral energy metabolism, glucose, L-lactate, and oxygen, consumed in a rat hippocampal dentate gyrus during a series of 5 s electrical stimulations of the perforant pathway at 2 min rest intervals. The amounts were calculated from the depth of each dip in the level of each substrate immediately following each stimulation [21] (see Figure 1). With every consecutive stimulation, beginning with the second one, the amount of glucose consumed decreased, while that of L-lactate increased. Oxygen consumption remained more or less constant.

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
