# Peer review of "Aerobic Glycolysis: A DeOxymoron of (Neuro)Biology"

_metabolites, 2022, doi:10.3390/metabo12010072_

Round 1

Reviewer 1 Report

General Comments:

      Not to diminish potential importance of this outstanding paper in any way, several suggestions are offered.  Neither the fault of the authors or others, it is sad that in science there exist silos of communication with something as important and essential understood in one field and completely unknown it other fields or even aspects of the same field.  Some suggestions are offered perhaps to broaden impact of the paper.  As well, the authors might decide, or not,  to add a section “The Brain and Beyond” to talk about the same issue in cardiology, environmental, muscle and exercise physiology. 

Specific Comments:

  1. The title is catchy and should serve well.But, could there be an addition to broaden?  “Aerobic Glycolysis: A Deoxymoron in Neurobiology and other Fields.”
  2. Abstract, line 12: The same problem exists in basic biology, biochemistry and physiology textbooks.Let’s face it, those who go into science and medicine are generally very smart people.  Hence, and errant concept learned in youth is hard to dislodge.  The authors write about neurobiology seasoned with examples from cancer research. Errant concepts are disadvantaging patient care is TBI, stroke, MI, heart failure, infection and sepsis to name a few. Hence the paper under consideration is very, very important with potential for wide translational impacts.
  3. Abstract, line 15: muscle and heart could be mentioned.
  4. Abstract, line 19: Postprandial and exercise metabolism could be mentioned.
  5. Introduction, line 38: “by many as real.”
  6. Introduction, lines 39-44: Surprisingly the paper of Hashimoto and Secher et al. (PMID: 29127193) is not cited. 
  7. Section 2, Page 3: Studies on humans are unmentioned.In addition to PMID 29127193, consider also PMID 29617642. 
  8. Section 3, Page 5, Lines 224-end: Bravo, I put a star near the text that might be highlighted in some way.Glycolytic flux leading to lactate production is high when OxPhos is high. They work together, hand in glove to sustain energy flux. As the authors point out, even in cancer lactate production and oxidative lactate disposal are present.  Some assert that there is a distinction between the oxygenated boundary of a tumor and its “anaerobic core,” but studies on cancer cell lines and mitochondria from those lines show oxidative lactate disposal. As well, the same thing happens in the beating human heart and working muscle.  See PMID: 33968972 for some citations if desired.  
  9. Section 3, Page 6, Line 240:  Cardiac metabolism comes to mind as another example of EC-coupling involving Na+/K+-ATPase.   In studies on intact humans as well as classic Langendorff rabbit hearts preparations (e.g., PMID: 173281) optimal contractility requires glucose.  FFAs and glycolysis cannot achieve that result; the conclusion being that there is some tissue compartment that requires glycolysis to sustain energy flux.  
  10. Line 319:Here, a big, bold summary header is suggested.
  11. And finally, a repeated suggestion and admonition.The paper is likely to have an impact, but it might be like spitting in an ocean of historic misunderstanding.  In addition to the above-mention suggestion to include a penultimate paragraph showing that the lessons learned are not unique to neuroscience or cancer research, it might be helpful to cite other recent works that try to make a similar point. Together, we might be able to make some waves.

Author Response

We appreciate the reviewer's comments and suggestions, which we believe help us to greatly improve the manuscript.                                     

  1. The title has been change somewhat to be more inclusive.
  2. Additions were made to the abstract to highlight the fact that the errant concept the manuscript addresses is more encompassing than in neurobiology.
  3. Other tissues are now mentioned in the abstract.
  4. Ditto for postprandial and exercise metabolism.
  5. Corrected.
  6. This paper is now cited.
  7. Ditto for this paper.
  8. This comment spurred us to add a line (268) in the manuscript and include three additional references (62-64). Thank you.
  9. This point is now covered by the previous point.
  10. A summary section (5) was added.
  11. In this newly added summary we included a paragraph highlighting the fact that the use of outdated, misleading terms has for-reaching consequences and new three references were included (71-73).

Reviewer 2 Report

This is an interesting opinion paper that points out some of the problems in the current views on brain metabolism in regard to the relative role of glucose and lactate. I have a few comments that the authors might wish to address if they so wish.

The authors rightly point out the lack of clarity of the term aerobic glycolysis. In the introduction, it might help the reader to understand the origin of the expression by pointing out that the opposition aerobic/anaerobic was originally applied to bacteria to describe the use of oxygen in energy production. Anaerobic generation of energy without glucose was thought to represent bacteria that existed before the earth atmosphere contained significant oxygen. The term was later used to describe muscle energy metabolism when oxygen becomes rate-limiting in exhaustive exercise.

In all cases, it was meant to delineate two forms of energy generation. It is quite clear however that the brain cannot function and survive very long without oxygen whereas it can when glucose levels are very low. The rapid brain death that follows asphyxiation pointed early researchers towards the idea that the brain mainly uses oxygen for energy production. Also possibly, the thought that such an evolved organ like the human brain would partially obtain its energy without oxygen like a primitive bacterium was not very appealing.    

One argument that could help the author’s thesis is that when blood levels of lactate are high, brain glucose is also elevated indicating a sparing of the transported glucose and a primacy of lactate use by the brain particularly in the active brain (Béland-Millar et al. 2017).

Another aspect not mentioned is the possibility that glucose uptake and possibly lactate uptake from blood to brain can be increased by upregulating transporters found at the BBB when local glucose needs are increased by local brain activity (Choeiri et al 2005).

Finally, one must remember that the measurements obtained by Hu and Wilson used electrochemical electrodes that cannot identify the flow direction of glucose or lactate. That means (in theory) that the minute decrease of lactate following electrical stimulation could be interpreted either as lactate being picked up by brain cells to be metabolized further or it could be lactate being transported out of the brain. Although I don’t subscribe to the latter, this limitation of electrode has to be pointed out. Also, the changes in glucose and lactate measured with electrodes also depends on how quickly glucose and lactate can be transported through the BBB. For example, the smaller dip in glucose (in comparison to lactate) observed following each electrical stimulation in Hu and Wilson could reflect the more efficient glucose transport that replenishes extracellular glucose more rapidly preventing a rapid fall. Conversely, if lactate transport out of the brain is more limited then it would tend to accumulate within the extracellular compartment. Finally lactate levels in the blood are much lower than glucose and that would influence the efficiency of the facilitated transport of glucose and lactate.

If I understood well, the author’s hypothesis is that L-lactate used in OXPHOS after the initial glycolysis step. Would the use of L-lactate in OXPHOS involves oxygen use? Again, if I understood correctly, that would not be compatible with the mismatch between glucose and oxygen use after brain stimulation originally described.  

Line 252: “mashed” is “meshed”

Author Response

We appreciate the reviewer's helpful comments. We have incorporated most of the reviewer's suggestions, while keeping in mind that our manuscript has meant to be response to a specific hypothesis paper that deals with aerobic glycolysis in the brain. Thus, we added an opening sentence to the Introduction section on the origin of the terms 'aerobic' and 'anaerobic.'

We agree with the reviewer's suggestion regarding glucose sparing due to lactate utilization and included a sentence to this effect (line 264) and the suggested reference (61).

A paragraph was included on the possible drawback of electrode use for tissue measurement of glucose and lactate (lines 128-132 and reference 33).

The use of lactate in oxphos is, of course, involves the consumption of oxygen. Unfortunately, in the original submitted manuscript an error was made in Figure 2, where the oxygen levels data presented were 1/10 of the correct values. That error is now corrected. Also. the reviewer's attention is called to the theoretical calculations of ATP production based on the measured glucose and lactate consumption measured by Hu & Wison (lines 158-182).

"Mashed" was corrected to "meshed."

Reviewer 3 Report

General:

Overall, I found the manuscript somewhat long and convoluted in explaining a relatively simple concept. Based on basic biochemistry, and objective evidence, the LDH reaction is near equilibrium. Lactate concentration is typically 10 or more times greater than pyruvate concentration. Therefore, lactate is the endproduct of glycolysis. In numerous physiological conditions, glycolysis is activated to a greater extent than are the aerobic pathways beginning with the NADH shuttles, pyruvate transport into mitochondria, and the pyruvate dehydrogenase reaction. Accordingly, regardless of the presence or absence of oxygen, lactate can accumulate. Essentially, there are two conditions that lead to lactate accumulation; accelerated glycolysis and an inadequate oxygen supply. Whether or not glycolysis is accelerated beyond the aerobic pathways during neural stimulation should be, and seems to be, the major focus of the manuscript.

The authors may also want to consider points against the notion of aerobic vs. anaerobic glycolysis in skeletal muscle as discussed in https://doi.org/10.1113/JP281335.

Efficiency is mentioned numerous times in the manuscript, but it’s never entirely clear what is exactly meant. Efficiency is most a ratio of two energy terms. A real calculation of efficiency of metabolic pathways is actually quite complicated. I recommend that the authors define very early on exactly what they mean when they use the term, efficiency or else use a different term. Certainly, we know that glycolysis begins with carbohydrate and degrades it incompletely; considerable energy remains in lactate. In this sense, the colloquial idea of "less efficient" is understandable. One could also argue that, absent oxygen utilization, glycolysis leaves 4+ ATPs on the table. Again, an exact definition or perhaps better, a different term, would be helpful.

L130-133 – This statement is confusing. It seems that you are denying the basic concept that lactate can, and often does, accumulate under conditions of ample oxygen supply. Is that what you mean? It certainly happens in skeletal muscle and cancer cells and likely many other cells as well, probably even brain. Or, are you arguing that it doesn’t happen in brain?

L143-147 – The idea that lactate can accumulate while at the same time, some pyruvate continues to be siphoned into the aerobic pathways doesn’t seem very surprising, based on studies from numerous tissue types, right? However, I think you want to propose that lactate is directly metabolized within the matrix of mitochondria - a concept that is controversial at best, and counter to the majority of evidence at worst.

Specific:

L39 - Why is it "thermodynamically defying"? There are explanations, yours being one.

L54 - It wouldn't be "bypassing" oxphos, simply not collecting the energy of the electrons in NADH and not being metabolized further.

L68 – Here and elsewhere, the authors should acknowledge that the majority of evidence (the authors have their own counter evidence) indicates that lactate is not directly oxidized by mitochondria without being first converted back to pyruvate somewhere outside the inner membrane of the mitochondria. Also, if one thinks the evidence is that mitochondria have LDH in the matrix and can metabolize lactate directly, a thermodynamic explanation is needed.

L78 – “labeled”

Figure 1 – This would be more informative if it presented absolute values rather than percentages.

L159 – Incorrect usage of the term, efficiency.

L161 and following – These data are suggestive of increased lactate uptake and increased oxygen uptake, but it should be noted that without blood flow, one cannot be absolutely certain.

L186-187 - Newsholme defined one ending of metabolic pathways as a nonequilibrium reaction. In that definition, glycolysis ends at the pyruvate dehydrogenase reaction because PDH is catalyzing a nonequilibrium reaction. Of course, there is glycolysis from glucose which may begin with glucose absorption in the GI tract, and glycolysis from glycogen which begins within cells that have glycogen stores. So, yes, by the Newsholme definition, glycolysis from glucose and glycolysis from glycogen are metabolic pathways. PDH is most likely its own pathway, then there’s the TCA cycle, etc.

L221-222 – It’s reasonable to conclude that any time the oxygen level is above zero and carbohydrate is the substrate, metabolism will continue beyond glycolysis, with or without additional lactate accumulation depending on the exact conditions.

L222-223 – See comment about L68.

L255-259- - This makes good sense.

L268-269 – Isn’t it likely that phosphocreatine to ATP is the first reaction that is activated most rapidly, and then glycolysis second?

L282-285 – This statement seems to run counter to your earlier statement (L242-246) about structural or localization constraints. If glycolysis is obligatory for the Na+/K+ pump, how can oxphos supply its ATP requirement?

L327-328 – Confusing sentence. Lactate can clearly accumulate even when oxygen is in ample supply. This sentence implies that is not the case.

Author Response

We appreciate the reviewer's comments and suggestions. The manuscript is mainly a response to a hypothesis paper published recently (Ref. 5), where the term 'aerobic glycolysis' is a term, which we find to be outdated and misleading. Based on all three reviewers' comments and suggestions we added several references and paragraphs to the manuscript in an effort to clarify several issues and to expand our views to include, as correctly suggested by this reviewer, to other tissues beside neuronal tissue.

Regarding the term 'efficiency' used in our manuscript, this is a term used in the hypothesis paper (5) to which our manuscript is responding. There, the authors declared glycolysis to be less efficient than oxphos, and rightfully so. Glycolysis leaves only 2 ATPs on the table, while oxphos, with lactate as its substrate, leaves 15 ATPs.

We absolutely agree that lactate can accumulate under conditions of ample oxygen supply. That is actually what Figure 1 exhibits and what, we believe, allows lactate to become the preferable oxphos substrate during brain stimulation and possibly during exercise. In addition, the ANLS hypothesis of Pellerin and Magistretti postulates lactate accumulation, while oxygen supply is abundant. Thus, lactate accumulation when oxygen is present occurs in the brain.

We do propose that lactate is directly metabolized within the mitochondrial matrix (see doi.org/10.3390/ijms222312620, reference 38 in the manuscript).

The term 'thermodynamically defying' relates to the hypothesis paper (5) to which our manuscript responds. There, the authors describes glycolysis as less efficient than oxphos. Accordingly, the preference for inferior efficiency is thermodynamically defying.

The last two points raised by the reviewer were already dealt with in our published papers (doi.org/10.3390/ijms222312620 and doi.org/10.3389/fnins.2018.00700).

Round 2

Reviewer 3 Report

No further comments.